# Late Recurrence in Breast Cancer: To Run after the Oxen or to Try to Close the Barn?

**DOI:** 10.3390/cancers13092026

**Published:** 2021-04-22

**Authors:** Romano Demicheli, Elia Biganzoli

**Affiliations:** Laboratory of Medical Statistics, Biometry and Epidemiology “Giulio A. Maccacaro”, Department of Clinical Sciences and Community Health & DSRC, University of Milan, Campus Cascina Rosa, Fondazione IRCCS Istituto Nazionale Tumori, 20133 Milano, Italy

**Keywords:** recurrence dynamics, tumor dormancy, cancer paradigms

## Abstract

**Simple Summary:**

The initial treatment of early breast cancer has achieved important clinical results over time. However, late recurrences after many years of disease-free survival remain an open question, which has recently attracted the attention of a few researchers. The authors of this commentary suggest that the approach emerging from scientific meetings regarding this subject is marred by the lack of attention to recent clinical and laboratory data. The role of tumor dormancy and the dynamics of disease recurrence are presented here and a more general reflection on therapeutic approaches to cancer is proposed.

**Abstract:**

The problem of late recurrence in breast cancer has recently gained attention and was also addressed in an international workshop held in Toronto (ON, Canada), in which several aspects of the question were examined. This Commentary offers a few considerations, which may be useful for the ongoing investigations. A few premises are discussed: (a) clinical recurrences, especially the late ones, imply periods of tumor dormancy; (b) a structured pattern of distant metastases appearance is detectable in both early and late follow-up times; (c) the current general paradigm underlying neoplastic treatments, i.e., that killing all cancer cells is the only way to control the disease, which is strictly sprouting from the somatic mutation theory, should be re-considered. Finally, a few research approaches are suggested.

## 1. Introduction

After a first encouraging phase associated to adjuvant chemotherapy and endocrine treatments [1,2], results of systemic therapy of breast cancer have successively slowed their initial impact on the disease prognosis. Therefore, during the 1990s, oncologists, following a quasi-Pavlovian reaction on the basis of the then current paradigm of cancer development, resolved to increase administered doses. This trendy line mimicked the behavior of the past surgeons who, always on the basis of the same paradigm, increased the extension of the surgery. This approach did not obtain meaningful improvements, as happened to surgeons as well [3]. Also, the therapeutic gains achieved more recently through innovative approaches have been deemed modest, limited to subsets of patients and burdened by toxicity, in spite of the efforts and resources mobilized [4].

This stalemate has prompted many clinical researchers to focus on more limited topics, identifying niche issues in which to carve out some specific therapeutic improvement. In this context, the question of late recurrence (i.e., local or distant recurrence occurring after 5 years of disease-free follow-up) in breast cancer has recently gained attention. It was also prompted by a meta-analysis focused on ER-positive breast cancer patients who were disease-free after 5 years of scheduled endocrine therapy [5]. Actually, several studies have shown that patients who are free from disease relapse at 5 years after first-line treatments (surgery plus adjuvant chemo/hormone therapy) remain at significant residual risk of relapse. The risk is dependent on several factors such as initial stage and tumor characteristics. For example, patients with stage I or II have a residual risk of 7% and 11%, respectively, while the analogous risks for patients with estrogen receptor negative and positive tumors are 7% and 13%, respectively [6]. Assessments can be refined by considering multiple prognostic parameters. For instance, for patients with ER-positive T1 disease the residual risk of distant recurrence was evaluated to be 13% with no lymph node involvement, 20% with one to three positive nodes, and 34% with four to nine nodes [5]. The histological grade among patients with T1 N0 disease changed the residual risk of distant recurrence that was estimated to be 10% for low-grade, 13% for intermediate-grade, and 17% for high-grade disease [5].

The subject of late recurrences was addressed in a focused international workshop held in Toronto (ON, Canada), in which several aspects of the question were examined [7,8]. Moreover, at the 2020 San Antonio Breast Cancer Symposium, an oral presentation in an educational session (E53) and a poster (PD9-11) further clarified research topics and clinical goals pursued in this area. The issue is quite important and calls into question both the research setting and the general philosophy underlying the proposed therapeutic modalities. We offer here a few considerations which we believe may be useful for the ongoing investigations.

## 2. Tumor Dormancy

A necessary premise from which to start is the recognition that clinical disease recurrences, especially the late ones, imply periods of tumor dormancy [9], a established notion in oncology [10]. This is not a foregone concept, when one considers that in the workshop there have been those (fortunately a minority) who were hypothesizing that late metastases originate from slow-growing indolent tumors [8] according to concepts of Halstedian memory by now largely superseded by the Fisherian vision [11]. It is curious to still hear such assumptions in spite of the fact that the relationship between clinical latency time and continuous tumor growth rate has been shown to be incompatible with experimental data and therefore abandoned several years ago [12]. In fact, the analysis of the size of local recurrences after mastectomy in relation to time to their clinical evidence, definitely clarified that they spent part of the time free from recurrence in a state of tumor dormancy. This finding profoundly changed the perspective of interpretation of the metastatic event and prompted reconsideration of tumor behavior during the follow-up. Indeed, information on tumor dormancy in humans may be inferred from the recurrence dynamics, since the lag-time between primary tumor removal and successive distant or local recurrence depends upon the state of metastatic microscopic foci at the time of surgery and on their growth pattern during the subclinical phase [13].

Actually, dormancy is a tumor state occurring at subclinical level, making it very difficult to conduct direct investigations on it in humans, beyond observations from bone marrow foci. In fact, what we know about tumor dormancy comes from experimental studies providing results, which are assumed to have validity also in humans. At least three mechanisms (cellular, angiogenic and immunological), are considered at the origin of cancer dormancy [14].

Cancer cell quiescence in G0–G1 arrest is the simplest dormancy condition observed, for instance, in spontaneous liver metastases from a murine mammary carcinoma [15], where dormant cells were also detected concurrently with progressively growing metastases. The G0–G1 arrest is induced by cell-microenvironment interaction, such as, for example, through urokinase plasminogen activator (uPA) binding to uPA receptor (uPAR) and the integrin α5β1, a complex acting upon ERK signaling.

Angiogenic dormancy, which was proposed by the Folkman J research group, relies on the observation that tumor size is strictly dependent on its vascularity, in the absence of which cannot exceed a limit diameter of about 1 mm [16]. In these avascular deposits, labelled micrometastases, cuffs are observable around pre-existent vessels a steady state is observable, with a high proliferation index, high apoptotic index and no necrosis [17]. The angiogenic dormancy may be interrupted by the so called “angiogenic switch”, when any cell of the focus (cancer cell or cancer stroma cell or others) acquires the angiogenic phenotype by secreting angiogenic factors (e.g., VEGF) or downregulating angiogenic suppressors (e.g., Angiostatin [18]).

The third mechanism of cancer dormancy involves the immune system within the old theory of immunosurveillance [19], which was recently restored and better detailed [20]. This theory hypothesizes that the immune system is able to constantly monitor the host tissues from the emergence of transformed cells. The immunological environment would select tumor variants that are more likely to survive in an immunocompetent host due to reduced immunogenicity or even because they have acquired evasion mechanisms. To describe the behavior of the immune system the new definition of “cancer immunoediting” has been proposed, including three phases: elimination, equilibrium, and “escape”. In the equilibrium phase immune system and tumor cells would enter a dynamic balance, in which the action on tumor cells would be sufficient to contain a number of tumor cells, which are genetically unstable. This condition of Darwinian selection is maintained until the emergence of a new population with reduced immunogenicity capable of growing without restriction. Such host-tumor condition has been realized in an experimental model where tumors may enter transiently into a period of equilibrium with the immune system and are assumed to mimic the equilibrium state by immune control [21]. Also, clinical clues to this equilibrium phase in humans surface from the transmission of cancer from transplant donors to recipients. A few observations suggest that pharmacological suppression of the immune system of these transplant recipients facilitates the growth of occult tumors of the transplanted organ. Such microscopic tumor foci are believed to have previously been maintained in equilibrium by the donor’s competent immune system [22].

## 3. Recurrence Dynamics

Another unavoidable point to be considered in breast cancer recurrences is the fact that recurrence appearance over time revealed [13,23] a very precise structure (confirmed in different databases), consisting of a series of successive peaks occurring at fixed times after the surgical treatment of the primary tumor. This regular distribution was subjected to careful analysis looking for different possible origins (artefact from data collection modalities, clustering effect from different anatomical metastatic sites or different types of patients). Moreover, its behavior was investigated by tumor and patient major characteristics (tumor size, axillary node involvement, estrogen receptor status, menopausal status, and others). It was concluded that the structure of recurrence dynamics should be considered an intrinsic trait of the metastatic process [13]. From the statistical viewpoint this situation is described as unexplained heterogeneity linked to a frailty concept, modelled according to a recent proposal [24]. On the basis of the tumor dormancy concept, the origin of this finding was explained by assuming the occurrence of a few dormancy states of the microscopic metastases, due to homeostatic restraints from the primary tumor. It is conceivable that this multiplicity of states is related to the recently discovered process of metastatic parallel progression. Contrary to traditional beliefs, data from laboratory and clinical studies are increasingly supporting the concept that disseminated tumor cells (DTCs) found in different tissues are not end products of the stepwise process of the primary evolutionary history, which spread late into the organism. On the contrary, DTCs would leave the primary early and would undergo further progression and metastatic growth at the seeded site [25,26]. Although poorly known, this phenomenon could concur to the multiplicity of dormancy states revealed by the clinical metastasis dynamics. Another phenomenon could be involved in the microscopic tumor foci development, the tumor self-seeding, a process by which circulating tumor cells colonize the primary from which they originated [27]. Indeed, as this process requires specific attractive signals from the primary, it is possible that it may also occur between metastatic foci, thus working against multiplicity. Our current knowledge is too limited to draw any conclusions on this topic.

What we can say is that following primary tumor surgical removal, dormant foci undergo a sudden wake up resulting in the acceleration of the metastatic process. Hence the process of synchronization, which originates the time distribution of clinical metastasis appearance [9]. The multi-peak dynamics of recurrences, which was also been found in other solid tumors we examined so far, was, therefore, interpreted as a manifestation of the different biological conditions of the microscopic tumor foci, shaping different durations of clinical dormancy [9,23], of the microscopic tumor foci. Regarding late recurrences, a structured pattern of distant metastases appearance was also detectable in patients who were disease-free at 5 years of follow-up [28]. This finding supported the view that tumor dormancy interruption was again related to some triggering and synchronizing event, which needs to be actively investigated. As noted above, the most obvious event correlating with microscopic metastasis awakening is primary tumor surgical removal [9,29] (which, however, remains an essential procedure), a fact that was recently experimentally confirmed in animal models [21]. This occurrence implies that what we see at a certain time of follow-up (e.g., at the tenth year) may have roots in what happened many years earlier, at the first line treatment, once again pointing out that cancer is fundamentally a systemic disease. Therefore, it is not unlikely that a further investigational effort may support this hypothesis, although other factors at the individual patient level might have some influence, such as concomitant diseases, immune system action or metabolic states. Yet, it is a little hard to accept that such individual random events alone originate a structured hazard rate pattern in a patient population.

## 4. Cell Kill, Dormancy and Cure

A further main concern is regarding the current general paradigm underlying neoplastic treatments. In our opinion, the still popular treatment approach continues to be closely related to somatic mutation theory (SMT) [30,31], implying that “once a cancer cell, always a cancer cell”, an axiom the corollary of which is the statement that killing all cancer cells is the only way to control the disease. However, an increasing number of experimental and clinical data have proven not to be compatible with SMT tenets.

Contrary to the underlying assumption of the SMT, gene alterations are not indispensable in carcinogenesis: inert substances such as metals, asbestos, plastics, if inserted into tissues of animal hosts generate tumors locally (foreign body carcinogenesis) without releasing genotoxic compounds capable of mutating the DNA of adjacent cells [32]. Moreover, the epithelio-centric view of tumorigenesis is clearly in conflict with findings proving evidence of the modulatory role of stroma [33] and in particular, with the discovery that the stroma may be a crucial target of the carcinogen [34]. The SMT addresses this point by an ad hoc explanation: incipient neoplasias would recruit and activate stromal cell types to assemble a specific stroma, reciprocally able to respond by enhancing the neoplastic phenotypes. That would be a strange ability for a “self-sufficient” genetically transformed cell!

Another critical point of SMT is the sentence “once a cancer cell, always a cancer cell”, which is unequivocal: irreversibility is inextricably linked to cancer. Yet, the experimental evidence contradicts this assumption, as cancer cells may sometimes reverse their “malignant” properties when placed among normal tissues [35]. The first demonstration of such phenomenon goes back to 1970 of the past century, with the normalization of teratocarcinoma cells transplanted into early blastocysts of mice, which resulted in viable offspring displaying a mosaic phenotype combining tissues derived from both the host’s normal cells and the grafted teratocarcinoma cells [36]. Confirmative observations were successively reported and specifically, it was observed in multiparous female rats that mammary gland stroma inhibits neoplastic development and induces the growth of normal ducts from the grafted tumor cells [37]. A well-known and often forgotten spontaneous clinic-pathologic regression should also be mentioned. Neuroblastoma, a childhood neoplasm that spontaneously regresses in at least 2% of cases [38], is probably the one with the highest documented rate of spontaneous regression [39]. In the regression process, the progressive transformation of neoplastic tissue into mature tissue of adulthood is observed [40]. Some kind of reversion process emerged also in the Swedish mammography screening program, where the cumulative incidence of invasive breast cancer was significantly higher in the screened group than it was in the control group. The assumption of neoplastic reversibility is consistent with the results of such data analysis, where it was established that “the natural course of many of the screen-detected invasive breast cancers is to spontaneously regress” [41]. Finally, the SMT is unable to justify the occurrence of host–cancer balance, which allows a few tumors to remain dormant for a lifetime while other similar tumors lethally grow fast [42,43].

## 5. A Change in Perspective

As a result of all of these SMT failures, new paradigms were proposed, which shifted the neoplastic process origin from the cell level to the tissue level, where cell-cell and cell-extracellular matrix relationships assume a central role in determining the architecture and evolution of the tissues [44,45]. In particular, it may be observed that tumors frequently behave as ‘organ-like structures’ [46].

Tissue-based cancer explanations assume that, although a role may be even attributed to deranged DNA coding sequences or epigenetic disorders of gene expression, cancer origin and behavior mainly relies on disturbed tissue interactions among cell populations. Therefore, a better knowledge of the relationship between tumor and host can identify therapeutic modalities, till now unknown because not actively researched. Or, to put it better, most investigations on such relationships are looking for selective cytotoxic means directed against alien transformed cells. This is the case of the research on the relationship of the immune system with the tumor that gave rise to treatments with immune checkpoints blockers [47]. This polarized view, induced by the SMT, is an approach substantially borrowed from the paradigms of bacterial infections [30,31] and runs the risk of blurring and hiding important sides of the disease versus host system relations. Indeed, such a reductive approach disregards the complexity of the homeostatic mechanisms emerging from the most recent researches on the integration between systems at the tissue level. An example is the overall picture of homeostatic balance in the intestine where the collaboration of immune cells, epithelial cells (enterocytes, entero-endocrine cells, tuft cells), neuro enteric system and microbiota is coming to light [48]. Here we are learning that these elements integrate different information derived from the nutritional, immune and whole organism status to pursue the complex task of the gut, where nutrient absorption, exclusion of pathogens and toxins and maintenance in the lumen of the guests are not infrequently in contrast between them. This approach goes beyond the traditional views and suggests a more complex picture of the immune system, where equilibrium and activity are modulated by the local conditions, the organismic context and the surrounding environment. In this respect, recent studies on mammary adipose tissue and symbiotic metabolic relationships with the immune system and tumor cells [49] seem to be going in a different and, in our opinion, more interesting direction.

A typical example of the poor attitude to investigate the homeostatic balances that are achieved at the neoplastic level is the very poor knowledge of transition modalities towards or out of tumor dormancy. Dormancy, on the contrary, is at present considered an obstacle to the current treatments that are mainly cytotoxic and need to hit not quiescent tumor foci. However, tumor dormancy could be a main goal of cancer treatments. Indeed, the presence of dormant tumor foci may already be a widely diffuse state among patients disease-free after standard treatments. Data were reported suggesting that subclinical tumor metastases are likely to be embedded in tissues in most long-term disease-free patients possibly staying dormant for the lifetime [42], thus practically making the long lasting tumor dormancy status superimposable to the tumor cure. We sincerely hope that investigations on tumor dormancy will soon get the maximum effort in terms of focused research and committed resources.

It is also desirable that oncological research gets rid of the conceptual approach outlined by the current warfare metaphors (enemy cells, attack, defence, destruction…) to turn to a more complex vision. We should stop promoting the illusion that the “magic bullet” is on the verge of being found, in the same way that the losing gambler continues to bet, dreaming that he will win the next hand. It is time to take other pathways and consider that a complex problem such as cancer, which escapes the understanding of the simplistic approach symbolized by war metaphors, can be approached in a more complex and “physiological” perspective. The possibility and usefulness of this approach is well suggested by the observation that a completely harmless and physiological/metabolic activity implemented by targeted physical exercise interventions may improve the disease outcomes, at least in certain subsets of patients [50,51]. This activity was delivered, akin to drug administration, during and beyond standard treatments in early breast cancer patients. Moreover, this improvement emerges after about 5 years following primary treatment [52], suggesting that physical activity may exert its long-term effects on dormant metastases during their subclinical development. The unexpected therapeutic efficacy of an “unarmed” practice, such as that of physical exercise is for us a last prop to a conviction matured in decades of research. Furthermore, broadening the discussion, knowledge of the effect of modifiable factors on breast cancer prognosis is very modest. There is a scarcity of studies in laboratory models regarding the impact of factors such as diet, circadian rhythms, and physical activity on disease progression. The same can be said on clinical studies where both modifiable risk factors and their impacts on long-term prognosis are understudied [53,54].

## 6. Conclusions

In our opinion, exploring the reasons underlying the metastatic long-term dripping is better than pursuing, yet with little success, their later killing. In other words, it is better to keep the barn of “dormancy” closed instead of trying to better kill the awakened and fleeing “oxen” cells. Of consequence, we suggest putting tumor dormancy in the focus of laboratory investigations and clinical studies with the aim to unveil, at different levels of complexity (e.g., cell metabolism, tissue cross talks, organism immune system level), possible homeostatic processes on which to intervene. Moreover, laboratory and clinical research, when collaborating to build up a reliable picture of breast cancer behavior according to the dormancy evidence, could give up, at least in part, on their monotonous routine (made, however, with the best intentions): the reductionist approach to cell investigation and the paranoid comparison of drug efficacy (more effective vs. less effective). Indeed, all these efforts, it must be remembered, are directed against proliferating foci and neglect the dormant ones, which are the main and long-lasting reasons for the disease recurrence.

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
