# Peer review of "Late Recurrence in Breast Cancer: To Run after the Oxen or to Try to Close the Barn?"

_cancers, 2021, doi:10.3390/cancers13092026_

Round 1
Reviewer 1 Report
This book provides useful information about the considerations of late recurrence of breast cancer research. I agree with the authors that interest in and research on cancer dormancy should proceed further. They underestimated the competition for drugs in the conclusion that, after cancer surgery, the oncologists tried to remove the dormant cancer cells through secondary chemotherapy or targeted therapy. It's a pity that after removing the aggressive cells, only the cells that can pass through the immune gate remain, resulting in late recurrence.
minor revision
1. Page 3, line 132: The mention that surgery is a factor that causes microscopic metastasis is misleading. In addition to the hidden malignant cells, surgical procedures for the removal of primary cancer are considered essential. Please write in a more refined expression.
Author Response
This book provides useful information about the considerations of late recurrence of breast cancer research. I agree with the authors that interest in and research on cancer dormancy should proceed further. They underestimated the competition for drugs in the conclusion that, after cancer surgery, the oncologists tried to remove the dormant cancer cells through secondary chemotherapy or targeted therapy. It's a pity that after removing the aggressive cells, only the cells that can pass through the immune gate remain, resulting in late recurrence.
Lines 464-468
….. could give up, at least in part, on their monotonous routine 390 (made, however, with the best intentions): the reductionist approach to cell investigation 391 and the paranoid comparison of drug efficacy (more effective vs. less effective). Indeed, 392 all these efforts, it must be remembered, are directed against proliferating foci and neglect 393 the dormant ones, which are the main and long-lasting reasons for the disease recurrence.
minor revision
- Page 3, line 132: The mention that surgery is a factor that causes microscopic metastasis is misleading. In addition to the hidden malignant cells, surgical procedures for the removal of primary cancer are considered essential. Please write in a more refined expression.
Line 276
… primary tumor surgical removal [9,29] (which, however, remains an essential procedure),
Reviewer 2 Report
The theories presented in the paper are intersting and thought provoking. Although they are well laid out, there are a number of run on sentences and individual words that appear to be direct translations from phrases or idioms in Italian but don’t work in english. A thorough proof read by a native english speaker would be very helpful. Examples: line 38 ‘gnaw’ would be better replaced by ‘carve out’; line 53 word ‘acquisition’ doesn’t make sense; line 103 ‘clinical clues....leak out...’, line 200 ‘quit to feed the illusion’ ; line 54 ‘some (a minority)” ; line 163 should be rephrased as ‘the 1970s’, there are also several places where the wrong preposition is used or one is missing eg line 157 able ‘to’ respond (to missing); line 12 need to delete ‘to’ in ‘to this subject’.
Author Response
The final text underwent the check of a native English-speaking colleague.
Reviewer 3 Report
Minor (non-mandatory) points: - 1. Introduction: I suggest to add some clinical data related to outcome of late recurrences compared to early recurrences. I also suggest to briefly discuss the phenomenom of early dissemination in breast cancer and topic of DTC (see. Klein CA Nat Rev Cancer. 2009 Apr;9(4):302-12. Hüsemann et al .Cancer Cell. 2008 Jan;13(1):58-68. - 3. Recurrence Dynamics: I suggest to briefly discuss data related to self-seeding at the end of first paragraph where authors discuss the relationship between tumor dormancy and primary tumor removal (Cell 139, 1315–1326 (2009). Breast Dis. 29, 27–36 (2008). - 4. Cell Kill, Dormancy and Cure: I suggest to add data describing spontaneous regression in early breast cancer, phenomenon, that is supported by screening (Zahl PH et al. Arch Intern Med. 2008 Nov 24;168(21):2311-6)Author Response
Minor (non-mandatory) points: -
- Introduction: I suggest to add some clinical data related to outcome of late recurrences compared to early recurrences.
Lines 43-73.
Actually, several studies have shown that patients who are free from disease relapse at 5 years after first-line treatments (surgery plus adjuvant chemo/hormone therapy) remain at significant residual risk of relapse. The risk is dependent on several factors such as initial stage and tumor characteristics. For example, patients with stage I or II have a residual risk of 7% and 11%, respectively, while the analogous risks for patients with estrogen receptor negative and positive tumors are 7% and 13%, respectively [6]. Assessments can be refined by considering multiple prognostic parameters. For instance, for patients with ER-positive T1 disease the residual risk of distant recurrence was evaluated to be 13% with no lymph node involvement, 20% with one to three positive nodes, and 34% with four to nine nodes [5]. The histological grade among patients with T1 N0 disease changed the residual risk of distant recurrence that was estimated to be 10% for low-grade, 13% for intermediate-grade, and 17% for high-grade disease [5].
- I also suggest to briefly discuss the phenomenom of early dissemination in breast cancer and topic of DTC (see. Klein CA Nat Rev Cancer. 2009 Apr;9(4):302-12. Hüsemann et al .Cancer Cell. 2008 Jan;13(1):58-68.
- Recurrence Dynamics: I suggest to briefly discuss data related to self-seeding at the end of first paragraph where authors discuss the relationship between tumor dormancy and primary tumor removal (Cell 139, 1315–1326 (2009). Breast Dis. 29, 27–36 (2008).
Lines 200-263.
It is conceivable that this multiplicity of states is related to the recently discovered process of metastatic parallel progression. Contrary to traditional beliefs, data from laboratory and clinical studies are increasingly supporting the concept that disseminated tumor cells (DTCs) found in different tissues are not end products of the stepwise process of the primary evolutionary history, which spread late into the organism. On the contrary, DTCs would leave the primary early and would undergo further progression and metastatic growth at the seeded site [25,26]. Although poorly known, this phenomenon could concur to the multiplicity of dormancy states revealed by the clinical metastasis dynamics. Another phenomenon could be involved in the microscopic tumor foci development, the tumor self-seeding, a process by which circulating tumor cells colonize the primary from which they originated [27]. Indeed, as this process requires specific attractive signals from the primary, it is possible that it may also occur between metastatic foci, thus working against multiplicity. Our current knowledge is too limited to draw any conclusions on this topic.
- Cell Kill, Dormancy and Cure: I suggest to add data describing spontaneous regression in early breast cancer, phenomenon, that is supported by screening (Zahl PH et al. Arch Intern Med. 2008 Nov 24;168(21):2311-6.
Lines 372-377.
Some kind of reversion process emerged also in the Swedish mammography screening program, where the cumulative incidence of invasive breast cancer was significantly higher in the screened group than it was in the control group. The assumption of neoplastic reversibility is consistent with the results of such data analysis, where it was established that “the natural course of many of the screen-detected invasive breast cancers is to spontaneously regress” [41].
Reviewer 4 Report
In this commentary, the authors bring forward some topics within the field of tumor recurrence where there is a need to explore in more depth to really understand the root cause and then develop therapeutics to prevent recurrence. Overall, the authors pose some interesting points and a perspective that emphasizes the need to improve how the field approaches recurrence research. Below I list the recommendation to address the weaknesses and to improve the overall message and increase the impact of this commentary.
Recommendations
Minor
- Many of the sentences are very long and there are grammatical errors which make the flow of reading very difficult. I recommend extensive editing on English language and style.
- Line 142 “4. Cell Kill, Dormancy and Cure” could be changed to “Cell death…”
- The title “Late Recurrence: To Run after the Oxen or to Try to Close the Barn? “makes an interesting point but there is no mention of the symbolism within the context of the arguments presented in the paper. I recommend that the authors circle back to this very interesting argument that they make with the title. It would really highlight and strengthen the story. I also suggest that the authors define “late” (ie late relative to what? What would “early” mean). Perhaps the authors should consider a different choice of word such as “long-latency” and "short-term latency" and specify that it is after a patient has been clinically declared disease-free because cancer cells are undetectable. This also raises the point that our diagnostic methods have limits of detection. Furthermore, it seems that this symbolism is alluding to a long-term maintenance of dormancy (the closed door) which in theory could be more effective than the ability to kill (chase the oxen) all the cancer cells (including the subpopulations that remain like cancer stem cells which give rise to recurrences or metastases).
Major
- Within the commentary, there is very little information about breast cancer specifically. As it stands, the commentary seems to be more general rather than specific to breast cancer. I recommend that the authors use some of the extensive literature from randomized clinical trials in which breast cancer recurrence estimates are derived. The authors should address the recurrence estimates amongst different risk groups (subtypes, TNM classifications). It would be valuable to address how the long-term latency impacts overall health outcomes in the patients and their eligibility for additional lines of treatment.
- It would be informative if the authors described some of the laboratory models (in vitro, in vivo, in silico) that are specifically used to address recurrence (not metastasis). From these models the authors should highlight how their assumptions and the clinical relevance (or lack thereof) impact the interpretation of what is currently known about recurrence.
- In lines 63-66, the authors state
“information on tumor dormancy in humans may be inferred from the recurrence dynamics, since the lag-time between primary tumor removal and successive recurrence depends upon the state of metastatic microscopic foci at the time of surgery and on their growth pattern during the subclinical phase.”
Are the cancer cells after tumor removal really metastatic foci? Or are they residual tumor cells that were not removed because they are undetectable? It is reasonable to speculate that both situations are possible, but the authors should acknowledge this and clarify this point.
- In lines 149-152 the authors describe how exogenous exposures have the ability to initiate tumorigenesis. This brings up the point that there are limited laboratory models that integrate modifiable risk factors (ie environmental exposures, physical activity, diet, circadian rhythm) in recurrence studies. Even in population-based studies the modifiable risk factors are understudied and there is limited long-term surveillance to address how they impact recurrence risk.
- Line 186-187 “between tumor and host can identify therapeutic modalities, till now unknown because not actively researched”
- Do the authors not consider the immune system a part of the host or contributing to the tissue interactions? I recommend the authors to clarify what they mean and list examples of the specific interactions that are “understudied”. In breast cancer there are extensive studies of the breast adipose tissue and the symbiotic metabolic relationships with the immune system and tumor cells. In short, if authors argue that this area is understudied, provide specific examples or comparisons.
Author Response
Many of the sentences are very long and there are grammatical errors which make the flow of reading very difficult. I recommend extensive editing on English language and style.
The final text underwent the check of a native English-speaking colleague.
Line 142 “4. Cell Kill, Dormancy and Cure” could be changed to “Cell death…”
We would prefer to maintain the word “kill” to emphasize the current paradigm of oncological treatments aimed at eliminating neoplastic cells.
The title “Late Recurrence: To Run after the Oxen or to Try to Close the Barn? “makes an interesting point but there is no mention of the symbolism within the context of the arguments presented in the paper. I recommend that the authors circle back to this very interesting argument that they make with the title. It would really highlight and strengthen the story.
Lines 456-459
In our opinion, exploring the reasons underlying the metastatic long-term dripping is better than pursuing, yet with little success, their later killing. In other words, it is better to keep the barn of "dormancy" closed instead of trying to better kill the awakened and fleeing "oxen" cells.
I also suggest that the authors define “late” (ie late relative to what? What would “early” mean). Perhaps the authors should consider a different choice of word such as “long-latency” and "short-term latency" and specify that it is after a patient has been clinically declared disease-free because cancer cells are undetectable.
Lines 40-41
the question of late recurrence (i.e. local or distant recurrence occurring 40 after 5 years of disease free follow-up)
This also raises the point that our diagnostic methods have limits of detection.
In our opinion, this subject would require too extensive a discussion that would divert attention from the central topic of the commentary.
Furthermore, it seems that this symbolism is alluding to a long-term maintenance of dormancy (the closed door) which in theory could be more effective than the ability to kill (chase the oxen) all the cancer cells (including the subpopulations that remain like cancer stem cells which give rise to recurrences or metastases).
See upper replay (lines 456-459)
Major
Within the commentary, there is very little information about breast cancer specifically. As it stands, the commentary seems to be more general rather than specific to breast cancer. I recommend that the authors use some of the extensive literature from randomized clinical trials in which breast cancer recurrence estimates are derived. The authors should address the recurrence estimates amongst different risk groups (subtypes, TNM classifications). It would be valuable to address how the long-term latency impacts overall health outcomes in the patients and their eligibility for additional lines of treatment.
Lines 43-73.
Actually, several studies have shown that patients who are free from disease relapse at 5 years after first-line treatments (surgery plus adjuvant chemo/hormone therapy) remain at significant residual risk of relapse. The risk is dependent on several factors such as initial stage and tumor characteristics. For example, patients with stage I or II have a residual risk of 7% and 11%, respectively, while the analogous risks for patients with estrogen receptor negative and positive tumors are 7% and 13%, respectively [6]. Assessments can be refined by considering multiple prognostic parameters. For instance, for patients with ER-positive T1 disease the residual risk of distant recurrence was evaluated to be 13% with no lymph node involvement, 20% with one to three positive nodes, and 34% with four to nine nodes [5]. The histological grade among patients with T1 N0 disease changed the residual risk of distant recurrence that was estimated to be 10% for low-grade, 13% for intermediate-grade, and 17% for high-grade disease [5].
It would be informative if the authors described some of the laboratory models (in vitro, in vivo, in silico) that are specifically used to address recurrence (not metastasis). From these models the authors should highlight how their assumptions and the clinical relevance (or lack thereof) impact the interpretation of what is currently known about recurrence. In lines 63-66, the authors state “information on tumor dormancy in humans may be inferred from the recurrence dynamics, since the lag-time between primary tumor removal and successive recurrence depends upon the state of metastatic microscopic foci at the time of surgery and on their growth pattern during the subclinical phase.” Are the cancer cells after tumor removal really metastatic foci? Or are they residual tumor cells that were not removed because they are undetectable? It is reasonable to speculate that both situations are possible, but the authors should acknowledge this and clarify this point.
We wonder if this points are derived from the term "recurrence" which does not refer, in the context of the manuscript, to local recurrence only. We have therefore specified by including "local recurrence" and "distant recurrence" (as synonyms for "metastasis") when needed (e.g., line 97: between primary tumor removal and successive distant or local recurrence).
In lines 149-152 the authors describe how exogenous exposures have the ability to initiate tumorigenesis. This brings up the point that there are limited laboratory models that integrate modifiable risk factors (ie environmental exposures, physical activity, diet, circadian rhythm) in recurrence studies. Even in population-based studies the modifiable risk factors are understudied and there is limited long-term surveillance to address how they impact recurrence risk.
Lines 449-454.
Furthermore, broadening the discussion, knowledge of the effect of modifiable factors on breast cancer prognosis is very modest. There is a scarcity of studies in laboratory models regarding the impact of factors such as diet, circadian rhythms, and physical activity on disease progression. The same can be said on clinical studies where both modifiable risk factors and their impacts on long-term prognosis are understudied [53,54].
Line 186-187 “between tumor and host can identify therapeutic modalities, till now unknown because not actively researched”. Do the authors not consider the immune system a part of the host or contributing to the tissue interactions? I recommend the authors to clarify what they mean and list examples of the specific interactions that are “understudied”. In breast cancer there are extensive studies of the breast adipose tissue and the symbiotic metabolic relationships with the immune system and tumor cells. In short, if authors argue that this area is understudied, provide specific examples or comparisons.
Lines 390-410.
Or, to put it better, most investigations on such relationships are looking for selective cytotoxic means directed against alien transformed cells. This is the case of the research on the relationship of the immune system with the tumor that gave rise to treatments with immune checkpoints blockers [47]. This polarized view, induced by the SMT, is an approach substantially borrowed from the paradigms of bacterial infections [30,31] and runs the risk of blurring and hiding important sides of the disease versus host system relations. Indeed, such a reductive approach disregards the complexity of the homeostatic mechanisms emerging from the most recent researches on the integration between systems at the tissue level. An example is the overall picture of homeostatic balance in the intestine where the collaboration of immune cells, epithelial cells (enterocytes, entero-endocrine cells, tuft cells), neuro enteric system and microbiota is coming to light [48]. Here we are learning that these elements integrate different information derived from the nutritional, immune and whole organism status to pursue the complex task of the gut, where nutrient absorption, exclusion of pathogens and toxins and maintenance in the lumen of the guests are not infrequently in contrast between them. This approach goes beyond the traditional views and suggests a more complex picture of the immune system, where equilibrium and activity are modulated by the local conditions, the organismic context and the surrounding environment. In this respect, recent studies on mammary adipose tissue and symbiotic metabolic relationships with the immune system and tumor cells [49] seem to be going in a different and, in our opinion, more interesting direction
This manuscript is a resubmission of an earlier submission. The following is a list of the peer review reports and author responses from that submission.